# Novel height estimation formula that accounts for the effects of aging based on lumbar length measurements in postmortem CT images

**Kotomi Kai, Midori Katsuyama, Takahito Hayashi** [iD]*

Department of Legal Medicine, Graduate School of Medical and Dental Sciences, Kagoshima University, Kagoshima, Japan

* takahito@m2.kufm.kagoshima-u.ac.jp

## Abstract

In forensic practice, personal identification using expert testimony is important in unidentified cases, and the height of the deceased is indispensable in identification. Although many height estimation formulas have been reported, height estimates are often too great in the elderly due to age-related shortening. In this study, we address this problem by developing a height estimation formula based on measurement of the lumbar spine, which is thought to shorten with age. To develop a height estimation equation based on lumbar spine length, 183 postmortem CT images taken at our institute from 2016 to 2023 (ages 19–95 years) were prepared for the training data-set and 78 images were used for the validation dataset. In all training dataset cases, anterior margin height (ALV), central height (CLV), and posterior margin height (PLV) of each lumbar vertebra and total lumbar spine length including the intervertebral disc (LVTL) were measured on 3D CT-reconstructed images. The sum of the ALV (SALV), CLV (SCLV), and PLV (SPLV) of all lumbar vertebrae were calculated by image analysis software, and the correlation between each index and height was examined. As a control, an estimation equation based on sternal length was developed. Significant positive correlations were observed between each of the lumbar spine indices and height, with the PLV of the second lumbar vertebrate (PLV2) (R=0.710), SPLV (R=0.762), and LVTL (R=0.761) showing the strongest correlation. $R^2$ and standard error of estimation (SEE) were 0.622 and 4.926 cm for PLV2, 0.683 and 4.515 cm for SPLV, and 0.692 and 4.448 cm for LVTL, respectively. Furthermore, estimation equations based on sternal length often estimated higher values for elderly persons and did not take into account the effect of aging, while those based on PVL2, SPLV, and LVTL showed no correlation with age. In conclusion, we consider that our new formula for estimating height based on lumbar spine length, especially on PLV2, SPLV, and LVTL, is not affected by aging.

**Data availability statement:** The data that support the findings of this study are uploaded to a trusted data repository (https://doi.org/10.5281/zenodo.17677851) and are fully available without restriction.

**Funding:** The author(s) received no specific funding for this work.

**Competing interests:** the authors have declared that no competing interests exist.

## Introduction

In a forensic autopsy, the diagnosis of cause of death, including that of an external cause of death, such as homicide, suicide, or accident, are indispensable items of expert testimony [1]. An unknown person can be identified by DNA polymorphism testing with samples from the person's lifetime or samples from blood relatives, prenatal dental findings, or fingerprints. In addition, personal characteristics such as gender, age, and height can lead to personal identification when provided to the investigative agency as information to search for the person in question [1–4]. However, an unidentified corpse that requires personal identification may be in poor condition, and, in many cases, corpses are severely damaged by accidents, postmortem damage by animals, burning, progressive postmortem changes, and skeletonization. The estimation of sex, age, and height of such corpses is based generally on the shape, length, and angle of various bones [1,5–8].

In the field of forensic medicine, many formulas for estimating height from the measurements of bones have been used [7,9,10]. The development of those estimation formulas are based on the actual bone lengths measured during autopsy, especially limb bone lengths. Recently, estimation formulas based on the measurements of various bones throughout the body, including the skull, sternum, pelvis, and spine, as well as limb bones, on postmortem CT images have been reported [11–14]. However, in Japan, the widely accepted height estimation formula in actual forensic practice remains based on the limb bone lengths reported in the mid-1900s [15–17]. Several issues have been described in the use of this height estimation formula based on limb bone length. First, body shapes have changed between the period when the estimation formulas were developed and the present day, and accurate height cannot be estimated by substituting the measured lengths of the bones of modern humans into the estimation formulas. Second, in many cases, limb bones do not remain completely intact due to burn damage, prenatal damage, or postmortem destruction, making estimation based on limb bone length difficult. Further, while height shortens with age [18–21], limb bones do not shorten with age, and, thus, estimates based on limb bone length often estimate the height of elderly persons to be greater than reality. In order to solve these problems, we examined postmortem CT images of contemporary subjects (1924–2001) [15–17], using the lumbar spine as the bone likely to remain after burning, damage, or destruction. Lumbar vertebrae are known to shorten in the vertical direction due to age-related changes in cancellous bone, which can lead to decreased bone density, compression fractures, and osteophyte formation [22–24]. Adding these examples to the sample should inform development of an estimation formula based on the measured length of the lumbar spine that takes into account the effect of aging. Simultaneously, we created an estimation formula based on the measured length of the sternum, another bone of the torso for which a height estimation formula has been reported [12,25–27], and compared the accuracy of height estimation between the two methods.

## Materials and methods

### Materials

Postmortem CT images of cadavers (131 males and 130 females) taken prior to autopsy at the Department of Legal Medicine, Graduate School of Medical and Dental Sciences, Kagoshima University between January 1, 2016 and December 31, 2023 were used as samples. Specimen exclusions included cases under 18 years old, lumbar vertebra counts other than 5, fractures other than morphological fractures, dislocations, and the presence of medical metal instruments such as screws and plates. Postmortem CT images of cadavers from October 9, 2019 to March 10, 2025 were accessed for this study. At the Department of Legal Medicine, postmortem CT imaging is performed routinely in all autopsy cases (200–250 cases/year), except in cases of completely skeletonized cadavers.

Autopsy cases used as samples ranged from 19 to 95 years old [males: mean age, 64.2±17.5 years (median, 67.0 years); females: mean age, 62.3±18.6 years (median, 66.5 years)] and in height from 136.0 to 174.5 cm [males: mean height, 163.3±6.8 cm (median, 165.0 cm); females: mean height, 153.8±5.8 cm (median, 153.5 cm)]. The average time interval between death and postmortem CT imaging was 4.8 days (range, 12 hours to 60 days), of which about 90% were within 7 days, excluding cases in which postmortem decomposition clearly made reading difficult.

The data from the cases were evaluated in the training (183 cases) and validation (78 cases) datasets for measurement of lumber spine length (S1 File). The training dataset included 92 males [mean age, 64.1±17.7 years (median, 68.0 years) and mean height, 162.9±6.9 cm (median, 164.5 cm)] and 91 females [mean age, 61.9±18.6 years (median, 66.0 years) and mean height, 153.7±6.1 cm (median, 153.0 cm)]. The validation dataset included 39 males [mean age, 64.5±17.1 years (median, 66.0 years) and mean height, 164.4±6.5 cm (median, 165.0 cm)] and 39 females [mean age, 63.4±18.9 years (median, 68.0 years) and mean height, 154.0±5.1 cm (median, 154.0 cm)]. Of the training dataset, 65 cases (35.5%) showed an old compression fracture in at least one lumbar vertebra, with 32 (17.5%) in the first lumbar vertebra (L1), 14 (7.7%) in L2, 19 (10.4%) in L3, 22 (12.0%) in L4, and 34 (18.6%) in L5. In addition, 13 cases (7.1%) had scoliosis, with 8 cases (4.4%) assessed as mild (Cobb angle, 11–24°), 5 (2.7%) as moderate (Cobb angle, 25–39°), and none as severe (Cobb angle, greater than 40°). Of the validation dataset, 26 cases (33.3%) showed an old compression fracture in at least one lumbar vertebra, with 12 (15.4%) in L1, 8 (10.1%) in L2, 6 (7.7%) in L3, 8 (6.7%) in L4, and 10 (12.8%) in L5. In addition, 2 cases (2.6%) had scoliosis, with 1 case (1.3%) assessed as mild, 1 (1.3%) as moderate, and none as severe.

Of data from 244 cases, of which 16 cases with sternal fractures were excluded from the 261 cases used for lumber spine measurements, the data from 171 cases and 73 cases were evaluated in the training and validation datasets for measurement of sternal length, respectively (S2 File). The training dataset included 85 males [mean age, 64.3±17.7 years (median, 67.0 years) and mean height, 162.9±7.0 cm (median, 164.0 cm)] and 86 females [mean age, 61.5±19.2 years (median, 66.0 years) and mean height, 154.1±5.8 cm (median, 154.0 cm)]. The validation dataset included 38 males [mean age, 64.3±17.4 years (median, 67.0 years) and mean height, 164.5±5.9 cm (median, 165.0 cm)] and 35 females [mean age, 61.5±17.5 years (median, 64.0 years) and mean height, 155.2±6.0 cm (median, 154.0 cm)]. The ratio of men to women in each age group was similar in all datasets (Fig 1).

### Measurement of lumbar spine length

We created three dimensional (3D) images using AZE Virtual Place CT image analysis software. Anterior margin height of lumbar vertebra (ALV), central height of lumbar vertebra (CLV), and posterior margin height of lumbar vertebra (PLV) were measured (Fig 2a, c, e), as described previously [28]. Lumbar vertebra total length (LVTL) was defined as the length from the upper anterior margin of the first lumbar vertebra (L1) to the lower anterior margin of the fifth lumbar vertebra (L5) (Fig 2b, d, f). Sum of ALV (SALV) was the sum of the anterior margin height of all lumbar vertebrae. Sum of CLV (SCLV) and Sum of PLV (SPLV) were calculated in the same manner as SALV. For all measurements, the average of three

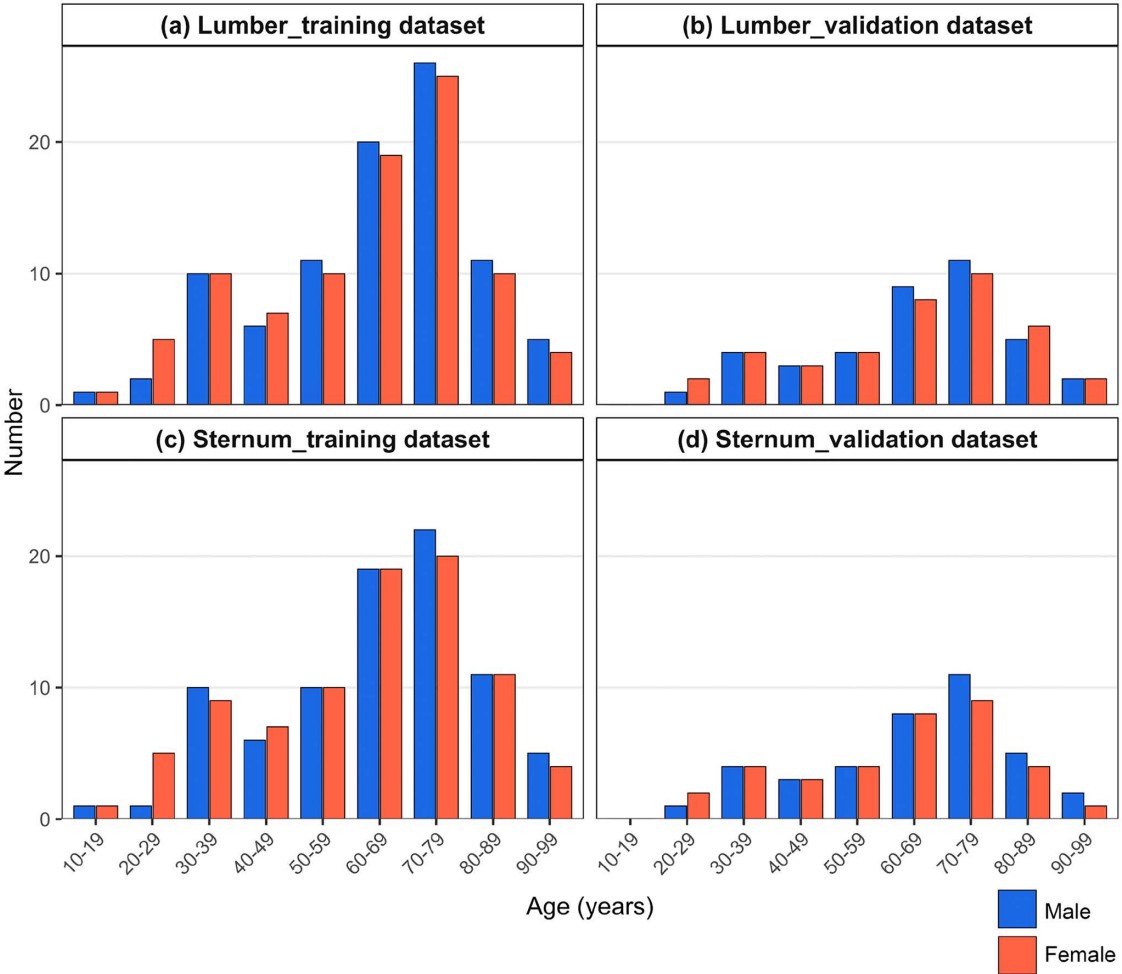

**Fig 1. Gender and age distribution of samples used in the training dataset (a) and validation dataset (b) of lumber spine measurement and in the training dataset (c) and validation dataset (d) of sternum measurement.**

measurements was taken as the measured value (intra-assessor reliability ICC(1, 3) was 0.999), and two persons performed the measurements separately (inter-assessor reliability ICC(1, 3) was 0.999).

## Measurement of sternal length

As in the measurement of lumbar spine length, a 3D reconstructed image was created, and the length from the jugular notch to the sternal angle and the length from the sternal angle to the lower end of the sternal body (excluding the xiphoid process) were measured. The combined length was defined as sternal length, as described previously [12]. An example of the measurement area is shown in Fig 2g. As with the lumbar vertebral length, two individuals took three separate measurements each.

## Statistical analysis

The data and related graphs were analyzed using R statistical software (version 4.5.1; The R Foundation) [29] and the significance level was set at less than 5%.

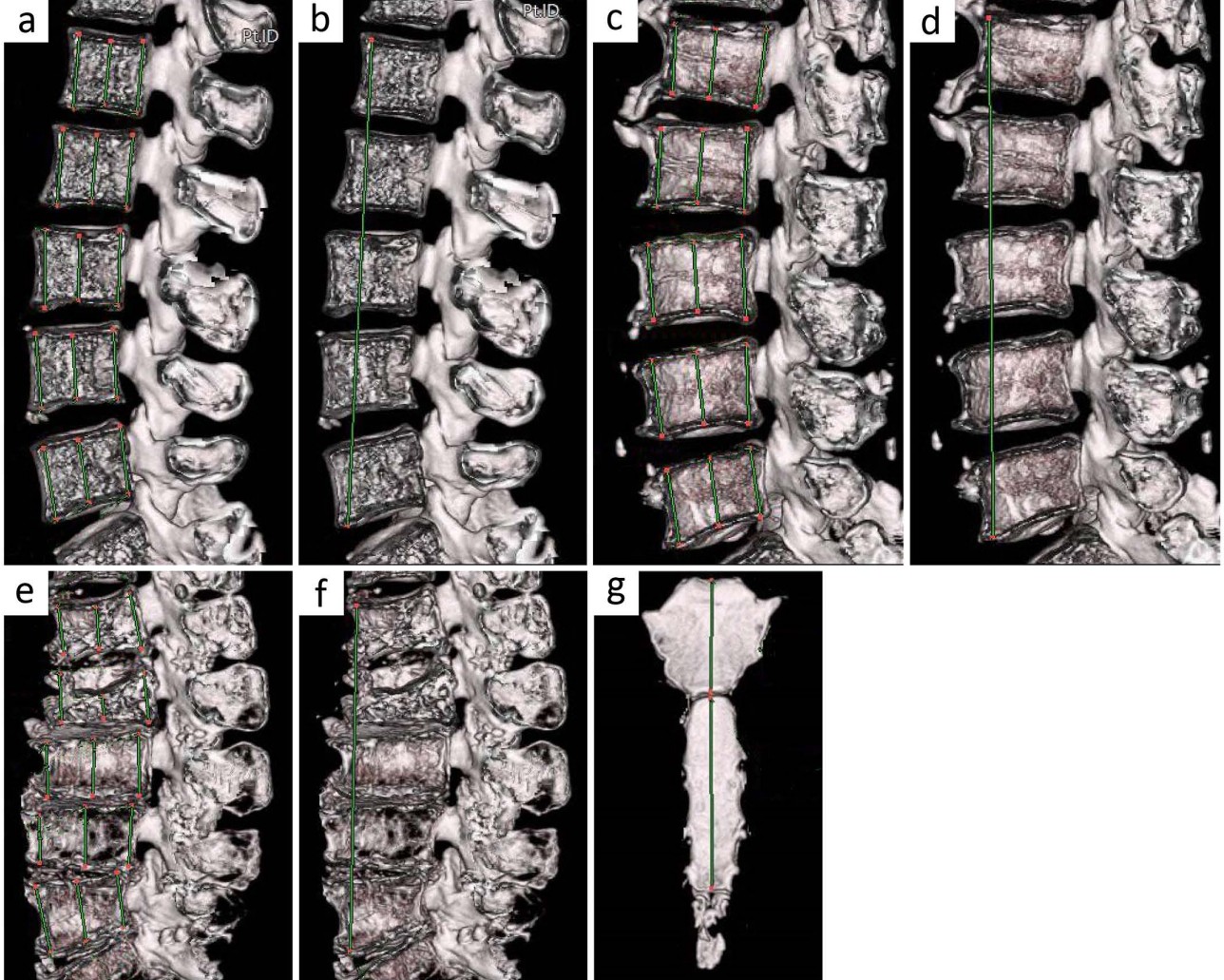

**Fig 2. Site of measurement of the lumbar spine and sternum.** The front, middle, and last lines of each lumbar vertebra indicate the ALV, CLV, and PLV measurement sites in an easily measurable, relatively young case (a), in a case of osteophyte formation in the lumbar vertebrae (c), and in a case with an old lumber compression fracture (e). (b) The line connecting the upper anterior margin of the first lumbar vertebra (L1) to the lower anterior margin of the fifth lumbar vertebra (L5) is the LVTL measurement site in same case as that in (a). The LVTL measurement sites in cases (d) and (e) are as indicated in (c) and (f), respectively. (g) The sum of the length from the jugular notch to the sternal angle and the length from the sternal angle to the lower edge of the sternal body is the measured length of the sternum.

The correlation between height and each parameter was evaluated for normality by the Shapiro-Wilk test, followed by Spearman's rank correlation coefficient. Since height distributions differed systematically by sex (Fig 3), sex was adjusted as a covariate in all models to avoid confounding. Analyses were conducted using multiple regression models with height as the dependent variable and each measurement item "m" and "Sex" as explanatory variables. Specifically, the following equation was applied individually for each measurement item, creating a total of 20 models (19 lumbar vertebrae, 1 sternum).

$$\text{Height} = \beta_0 + \beta_1 \, m + \beta_2 \, \text{Sex} \, (\text{Male} = 1, \text{Female} = 0)$$

Regression analysis was used to calculate regression equations for the height estimation equations based on each parameter. An analysis on residuals was performed as a precondition for linear regression. First, the normality of the

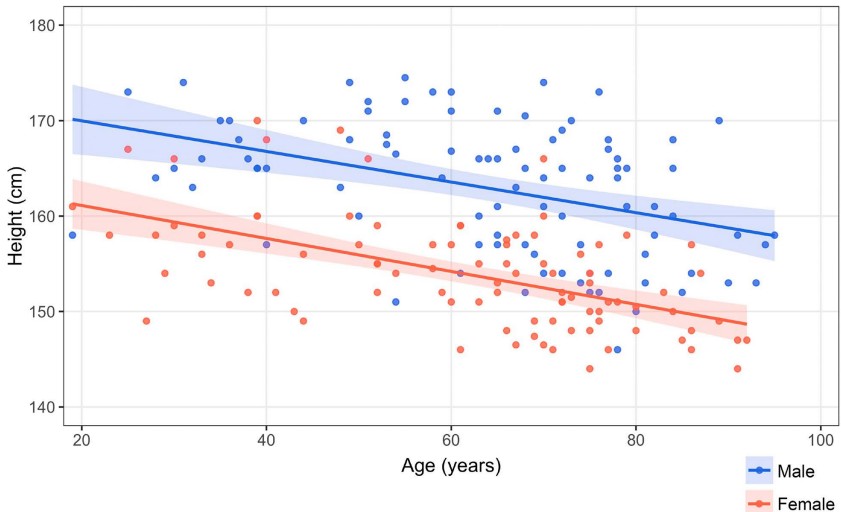

**Fig 3. Scatter plot showing the relationship between age and height by gender. The distribution of height differs systematically by gender.**

residuals was evaluated with the Shapiro-Wilk test. Next, the independence and equal-variance of the residuals were confirmed by creating a scatter plot of the residuals and estimated height. Finally, the linearity of the residuals was confirmed by creating a scatter plot of the relationship between the standardized residuals and the standardized predictions. The "coefficient of determination adjusted for the degree of freedom ($R^2$)" and "standard error of estimation (SEE, cm) were calculated in the multiple regression to evaluate the significance of the regression. Moreover, correlation analysis was used to examine whether the height estimation equation was affected by age. The height estimation formulas derived from the regression analyses were verified using validation datasets. Bias was calculated as the mean error (ME) and inaccuracy as the mean absolute error (MAE), as described previously [30].

## Ethics statement

This study was approved by the Ethics Committee on Epidemiological and its related Studies, Sakuragaoka Campus, Kagoshima University (Approval No190149-Epi. modif.4 and 5.) and was carried out in accordance with the principles of the Declaration of Helsinki. Moreover, this study was conducted using autopsy records, and we were unable to obtain informed consent from the bereaved family for the use of the records. Therefore, in accordance with the "Ethical Guidelines for Medical Research Involving Human Subjects (enacted by the Ministry of Health, Labor and Welfare in Japan)," Section 12−1 (2) (a) (c), information on the implementation of the study was posted on our website (http://www.kufm.kagoshima-u.ac.jp/~legalmed/), and if there was a request to refuse the use of samples for research, the specific samples were excluded from the study. In addition, the Ethics Committee on Epidemiological and its related Studies, Sakuragaoka Campus, Kagoshima University (Approval No190149-Epi. modif.4.) has approved the waiver for the informed consent of this study, as in our previous studies on other human samples [31].

## Results

### Correlation between height and each measurement

Descriptive statistics for all subject parameters in the training dataset are shown in Tables 1 and 2. Shapiro-Wilk's test showed that all lumbar spine measurements, except PLV2 (number indicates specific lumbar vertebrate), PLV3, PLV5, SALV, SPLV, and LVTL and sternum measurement did not follow a normal distribution. Therefore, Spearman's rank

**Table 1. Lumber spine measurements in the training dataset and validation dataset.**

| | Mean | Median | SD | Mean | Median | SD | Mean | Median | SD |
|---|---|---|---|---|---|---|---|---|---|
| Training dataset | | | | | | | | | |
| | All subjects (n = 183) | | | Male (n = 92) | | | Female (n = 91) | | |
| ALV1 (mm) | 22.8 | 23.5 | 3.5 | 22.8 | 23.5 | 3.6 | 22.7 | 23.5 | 3.6 |
| CLV1 (mm) | 22.1 | 23.1 | 4.5 | 22.9 | 23.8 | 4.4 | 21.3 | 22.4 | 4.4 |
| PLV1 (mm) | 26.5 | 26.6 | 2.3 | 27.3 | 27.7 | 2.2 | 25.7 | 25.8 | 2.2 |
| ALV2 (mm) | 24.9 | 25.1 | 2.2 | 25.0 | 25.2 | 2.0 | 24.7 | 25.1 | 2.3 |
| CLV2 (mm) | 23.3 | 23.6 | 3.0 | 24.1 | 24.2 | 2.4 | 22.5 | 23.4 | 3.2 |
| PLV2 (mm) | 26.9 | 26.7 | 2.0 | 27.6 | 27.6 | 1.9 | 26.1 | 26.1 | 1.8 |
| ALV3 (mm) | 25.8 | 25.9 | 2.2 | 25.8 | 25.5 | 2.3 | 25.9 | 26.2 | 2.2 |
| CLV3 (mm) | 23.1 | 23.2 | 3.0 | 23.7 | 23.5 | 2.3 | 22.4 | 23.1 | 3.5 |
| PLV3 (mm) | 26.4 | 26.3 | 2.1 | 27.1 | 27.0 | 2.1 | 25.8 | 25.9 | 1.8 |
| ALV4 (mm) | 25.5 | 25.6 | 2.3 | 26.0 | 25.9 | 2.0 | 25.1 | 25.3 | 2.4 |
| CLV4 (mm) | 22.3 | 22.7 | 2.9 | 23.2 | 23.2 | 2.0 | 21.5 | 22.4 | 3.5 |
| PLV4 (mm) | 24.7 | 24.8 | 2.2 | 25.5 | 25.5 | 2.0 | 23.9 | 24.2 | 2.1 |
| ALV5 (mm) | 25.6 | 25.8 | 2.3 | 26.2 | 26.2 | 1.8 | 25.1 | 25.6 | 2.6 |
| CLV5 (mm) | 21.9 | 22.2 | 2.7 | 22.7 | 23.1 | 2.2 | 21.0 | 21.7 | 2.9 |
| PLV5 (mm) | 22.7 | 22.7 | 2.1 | 23.2 | 23.3 | 2.1 | 22.1 | 22.3 | 2.0 |
| SALV (mm) | 124.6 | 125.3 | 9.6 | 125.8 | 125.7 | 9.0 | 123.4 | 124.3 | 10.2 |
| SCLV (mm) | 112.6 | 114.7 | 13.6 | 116.6 | 116.9 | 11.3 | 108.7 | 112.6 | 14.6 |
| SPLV (mm) | 127.2 | 126.9 | 9.4 | 130.7 | 130.4 | 9.0 | 123.6 | 124.9 | 8.3 |
| LVTL (mm) | 163.3 | 165.3 | 12.5 | 167.2 | 167.5 | 12.6 | 159.4 | 161.3 | 11.3 |
| Validation dataset | | | | | | | | | |
| | All subjects (n = 78) | | | Male (n = 39) | | | Female (n = 39) | | |
| ALV1 (mm) | 23.1 | 23.5 | 2.3 | 23.5 | 23.7 | 1.8 | 22.7 | 23.1 | 2.8 |
| CLV1 (mm) | 22.5 | 23.3 | 3.5 | 23.6 | 24.0 | 2.5 | 21.4 | 22.6 | 4.0 |
| PLV1 (mm) | 26.5 | 26.6 | 1.8 | 27.4 | 27.4 | 1.5 | 25.7 | 25.8 | 1.8 |
| ALV2 (mm) | 24.7 | 25.1 | 2.1 | 25.1 | 25.4 | 1.7 | 24.4 | 24.4 | 2.3 |
| CLV2 (mm) | 23.0 | 23.4 | 3.4 | 24.1 | 24.2 | 2.0 | 21.9 | 22.9 | 4.1 |
| PLV2 (mm) | 26.7 | 26.7 | 2.1 | 27.7 | 27.3 | 1.8 | 25.7 | 26.1 | 2.0 |
| ALV3 (mm) | 25.6 | 25.8 | 2.1 | 26.2 | 26.1 | 1.4 | 25.1 | 25.6 | 2.5 |
| CLV3 (mm) | 23.0 | 23.2 | 3.0 | 23.9 | 23.9 | 1.7 | 22.1 | 22.6 | 3.7 |
| PLV3 (mm) | 26.4 | 26.1 | 1.9 | 27.2 | 26.8 | 1.9 | 25.7 | 25.8 | 1.7 |
| ALV4 (mm) | 25.7 | 25.9 | 2.1 | 25.8 | 26.2 | 2.1 | 25.5 | 25.6 | 2.1 |
| CLV4 (mm) | 22.7 | 22.9 | 2.7 | 23.3 | 23.6 | 2.6 | 22.1 | 22.7 | 2.7 |
| PLV4 (mm) | 24.9 | 24.7 | 2.1 | 25.6 | 25.6 | 1.9 | 24.1 | 24.3 | 2.0 |
| ALV5 (mm) | 26.1 | 26.1 | 1.7 | 26.6 | 26.7 | 1.7 | 25.7 | 25.9 | 1.7 |
| CLV5 (mm) | 22.0 | 22.3 | 2.9 | 22.4 | 23.0 | 3.0 | 21.6 | 22.0 | 2.7 |
| PLV5 (mm) | 22.9 | 22.7 | 2.0 | 23.3 | 22.9 | 1.8 | 22.5 | 22.6 | 2.1 |
| SALV (mm) | 125.3 | 125.6 | 7.8 | 127.2 | 127.2 | 6.1 | 123.4 | 122.7 | 8.9 |
| SCLV (mm) | 113.1 | 114.5 | 12.7 | 117.3 | 117.5 | 8.9 | 109.0 | 112.4 | 14.5 |
| SPLV (mm) | 127.4 | 126.9 | 8.8 | 131.1 | 130.0 | 7.4 | 123.7 | 125.1 | 8.5 |
| LVTL (mm) | 165.5 | 167.1 | 10.2 | 169.6 | 169.9 | 8.1 | 161.5 | 164.6 | 10.6 |

SD, standard deviation; ALV, anterior margin height of lumber vertebra; CLV, central height of lumber vertebra; PLV, posterior margin height of lumber vertebra; SALV, the sum of ALV; SCLV, the sum of CLV; SPLV, the sum of PLV; LVTL, the lumbar vertebra total length. The specific lumbar vertebrate number is indicated in each dataset measurement.

**Table 2. Sternal bone measurements in the training dataset and validation dataset.**

|  | Mean | Median | SD | Mean | Median | SD | Mean | Median | SD |
|---|---|---|---|---|---|---|---|---|---|
| **Training dataset** | | | | | | | | | |
|  | All subjects (n = 171) | | | Male (n = 85) | | | Female (n = 86) | | |
| SB (mm) | 137.6 | 135.2 | 14.7 | 147.6 | 146.8 | 12.5 | 127.6 | 126.8 | 8.8 |
| **Validation dataset** | | | | | | | | | |
|  | All subjects (n = 73) | | | Male (n = 38) | | | Female (n = 35) | | |
| SB (mm) | 140.1 | 141.1 | 14.3 | 149.2 | 148.5 | 11.4 | 130.3 | 128.8 | 10.1 |

SD, standard deviation; SB, sternal bone

correlation coefficient was applied to the correlation between height and each measured value because at least one of the data did not follow a normal distribution. Fig 4 shows a scatter plot of each measurement versus height. There was a significant positive correlation between each of the lumbar spine measurements and height (Table 3). Among them, PLV1 (R = 0.723), PLV2 (R = 0.710), SPLV (R = 0.762), and LVTL (R = 0.761) showed a strong correlation with height. On the other hand, ALV1 (R = 0.462) and ALV3 (R = 0.408) showed a weak correlation with height. For all lumbar vertebrae,

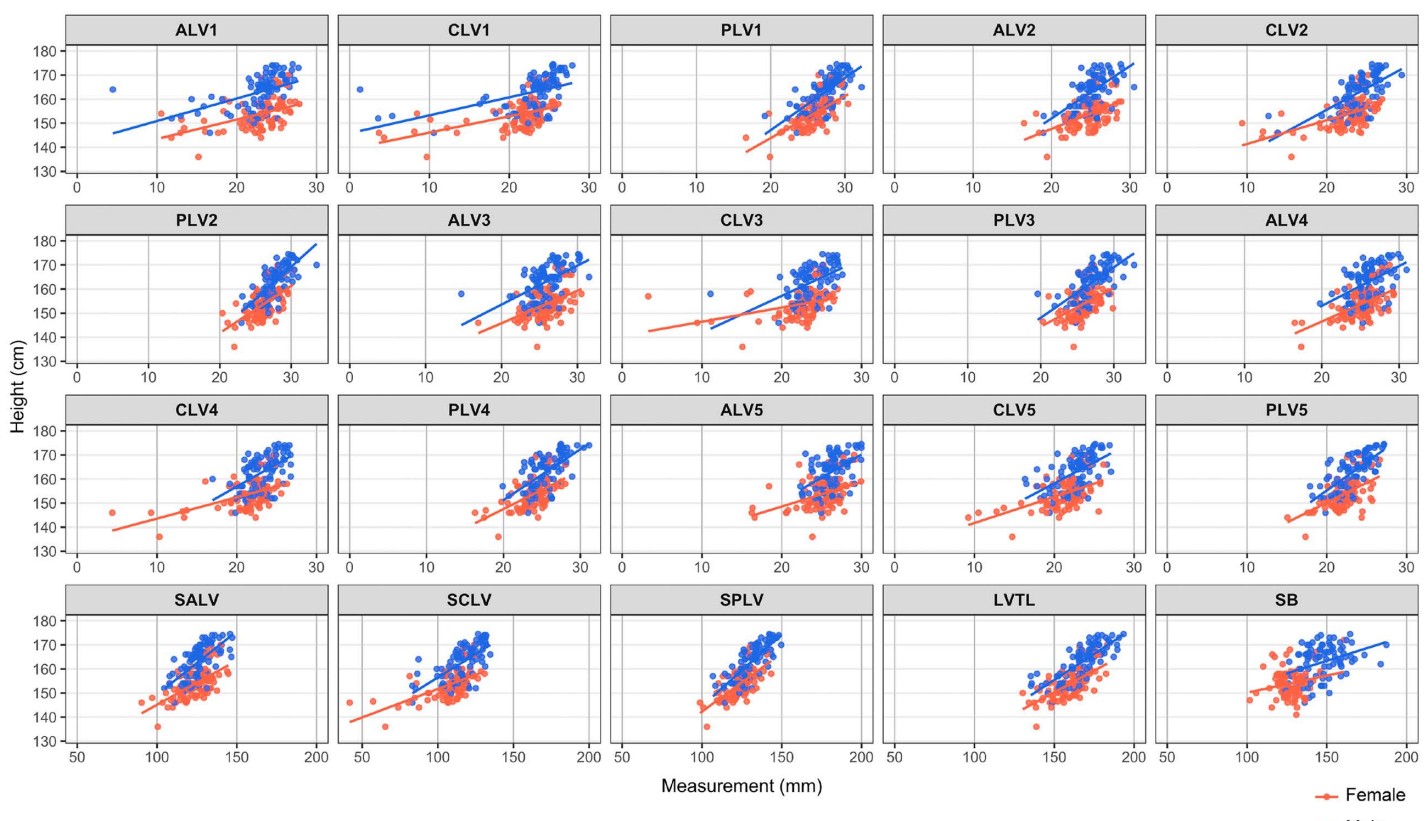

**Fig 4. Scatter plot showing the relationship between each bone measurement and height by gender.** ALV, anterior margin height of lumber vertebra; CLV, central height of lumber vertebra; PLV, posterior margin height of lumber vertebra; SALV, the sum of ALV; SCLV, the sum of CLV; SPLV, the sum of PLV; LVTL, the lumbar vertebra total length; SB, sternal bone. The specific lumbar vertebrate number is indicated in each dataset measurement.

**Table 3. Correlation coefficient between height and each parameter.**

| | All subjects | | Males | | Females | |
|---|---|---|---|---|---|---|
| | R | p-value | R | p-value | R | p-value |
| ALV1 | 0.462 | <0.01 | 0.630 | <0.01 | 0.540 | <0.01 |
| CLV1 | 0.622 | <0.01 | 0.624 | <0.01 | 0.516 | <0.01 |
| PLV1 | 0.723 | <0.01 | 0.732 | <0.01 | 0.621 | <0.01 |
| ALV2 | 0.465 | <0.01 | 0.608 | <0.01 | 0.464 | <0.01 |
| CLV2 | 0.623 | <0.01 | 0.637 | <0.01 | 0.588 | <0.01 |
| PLV2 | 0.710 | <0.01 | 0.720 | <0.01 | 0.568 | <0.01 |
| ALV3 | 0.408 | <0.01 | 0.600 | <0.01 | 0.489 | <0.01 |
| CLV3 | 0.533 | <0.01 | 0.581 | <0.01 | 0.486 | <0.01 |
| PLV3 | 0.642 | <0.01 | 0.686 | <0.01 | 0.542 | <0.01 |
| ALV4 | 0.491 | <0.01 | 0.489 | <0.01 | 0.504 | <0.01 |
| CLV4 | 0.553 | <0.01 | 0.544 | <0.01 | 0.480 | <0.01 |
| PLV4 | 0.630 | <0.01 | 0.606 | <0.01 | 0.562 | <0.01 |
| ALV5 | 0.488 | <0.01 | 0.504 | <0.01 | 0.428 | <0.01 |
| CLV5 | 0.609 | <0.01 | 0.593 | <0.01 | 0.483 | <0.01 |
| PLV5 | 0.622 | <0.01 | 0.706 | <0.01 | 0.489 | <0.01 |
| SALV | 0.566 | <0.01 | 0.704 | <0.01 | 0.607 | <0.01 |
| SCLV | 0.653 | <0.01 | 0.653 | <0.01 | 0.580 | <0.01 |
| SPLV | 0.762 | <0.01 | 0.792 | <0.01 | 0.665 | <0.01 |
| LVTL | 0.761 | <0.01 | 0.806 | <0.01 | 0.664 | <0.01 |
| SB | 0.575 | <0.01 | 0.349 | <0.01 | 0.060 | 0.582 |

R, Spearman's rank correlation coefficient; ALV, anterior margin height of lumber vertebra; CLV, central height of lumber vertebra; PLV, posterior margin height of lumber vertebra; SALV, the sum of ALV; SCLV, the sum of CLV; SPLV, the sum of PLV; LVTL, the lumbar vertebra total length; SB, sternal bone. The specific lumbar vertebrate number is indicated in each dataset measurement.

the correlation was stronger for posterior border height than anterior border or central height. The correlation coefficient between sternal length and height was 0.575, indicating a moderate correlation.

## Height estimation formula

Table 4 shows the height estimation formula, coefficient of determination adjusted for the degree of freedom ($R^2$), and standard error of estimation (SEE) derived from the multiple regression analyses. The $R^2$ were 0.692 for LVTL, 0.683 for SPLV, 0.625 for PLV1, and 0.622 for PLV2, and their SEE were 4.448 cm for LVTL, 4.515 cm for SPLV, 4.911 cm for PLV1, and 4.926 cm for PLV2, with LVTL having the lowest SEE. Estimation formulas were calculated as LVTL, y = 89.881 + 0.400*(LVTL) + 6.065*(Sex); SPLV, y = 86.328 + 0.545*(SPLV) + 5.276*(Sex); PLV1, y = 103.435 + 1.960*(PLV1) + 6.000*(Sex); and PLV2, y = 92.922 + 2.328*(PLV2) + 5.666*(Sex). On the other hand, $R^2$ and SEE for SB were 0.389 and 6.119 cm, respectively, all of which had smaller $R^2$ and larger SEE compared to LVTL, SPLV, PVL1, and PLV2. Furthermore, the residuals obtained for all estimation formulas are normally distributed (Table 5) and independent (Fig. 5), exhibit constant variance (homoscedasticity) (Fig. 5), and demonstrate linearity between the predictor and response variables (Fig. 6).

## Validation test

Table 6 summarizes the results of the validation tests. When applying the height estimation formula from the regression analysis of lumber spine measurement and sternal bone measurement to the validation dataset, the results were

**Table 4. Height estimation formula from multiple regression analysis.**

| | Regression formula | R² | SEE (cm) | p-value |
|---|---|---|---|---|
| ALV1 | y = 133.913 + 0.872*(ALV1) + 9.100*(Sex) | 0.483 | 5.761 | <0.01 |
| CLV1 | y = 138.581 + 0.712*(CLV1) + 8.048*(Sex) | 0.489 | 5.728 | <0.01 |
| PLV1 | y = 103.435 + 1.960*(PLV1) + 6.000*(Sex) | 0.625 | 4.911 | <0.01 |
| ALV2 | y = 112.130 + 1.684*(ALV2) + 8.648*(Sex) | 0.542 | 5.427 | <0.01 |
| CLV2 | y = 124.884 + 1.282*(CLV2) + 7.183*(Sex) | 0.543 | 5.420 | <0.01 |
| PLV2 | y = 92.922 + 2.328*(PLV2) + 5.666*(Sex) | 0.622 | 4.926 | <0.01 |
| ALV3 | y = 115.171 + 1.490*(ALV3) + 9.316*(Sex) | 0.509 | 5.619 | <0.01 |
| CLV3 | y = 134.055 + 0.879*(CLV3) + 7.987*(Sex) | 0.439 | 6.002 | <0.01 |
| PLV3 | y = 105.003 + 1.890*(PLV3) + 6.764*(Sex) | 0.553 | 5.357 | <0.01 |
| ALV4 | y = 115.707 + 1.516*(ALV4) + 7.824*(Sex) | 0.510 | 5.612 | <0.01 |
| CLV4 | y = 129.433 + 1.132*(CLV4) + 7.183*(Sex) | 0.492 | 5.712 | <0.01 |
| PLV4 | y = 109.788 + 1.840*(PLV4) + 6.162*(Sex) | 0.561 | 5.311 | <0.01 |
| ALV5 | y = 120.907 + 1.310*(ALV5) + 7.670*(Sex) | 0.464 | 5.870 | <0.01 |
| CLV5 | y = 125.616 + 1.336*(CLV5) + 6.988*(Sex) | 0.521 | 5.548 | <0.01 |
| PLV5 | y = 108.881 + 2.026*(PLV5) + 6.930*(Sex) | 0.592 | 5.120 | <0.01 |
| SALV | y = 99.835 + 0.437*(SALV) + 8.151*(Sex) | 0.608 | 5.017 | <0.01 |
| SCLV | y = 121.794 + 0.294*(SCLV) + 6.857*(Sex) | 0.564 | 5.294 | <0.01 |
| SPLV | y = 86.328 + 0.545*(SPLV) + 5.276*(Sex) | 0.683 | 4.515 | <0.01 |
| LVTL | y = 89.881 + 0.400*(LVTL) + 6.065*(Sex) | 0.692 | 4.448 | <0.01 |
| SB | y = 130.268 + 0.186*(SB) + 5.097*(Sex) | 0.389 | 6.119 | <0.01 |

$R^2$, coefficient of determination adjusted for the degree of freedom; SEE, standard error of estimation; Sex (male = 1, Female = 0); Measurements for ALV1 to SB are in mm; ALV, anterior margin height of lumber vertebra; CLV, central height of lumber vertebra; PLV, posterior margin height of lumber vertebra; SALV, the sum of ALV; SCLV, the sum of CLV; SPLV, the sum of PLV; LVTL, the lumbar vertebra total length; SB, sternal bone. The specific lumbar vertebrate number is indicated in each dataset measurement.

−1.277~0.016 cm and 3.653~4.413 cm for ME and MAE, respectively. The ME were −0.753 cm for PLV1, −1.277 cm for PVL2, −0.747 cm for SPLV, and 0.016 cm for VLTL. The MAE were 3.734 cm for PLV1, 4.066 cm for PVL2, 3.819 cm for SPLV, and 3.712 cm for VLTL.

### The effect of age on height estimation formula

To examine the effect of age on height estimation formulas, we performed correlation analysis on the relationship between age and residuals obtained by applying the validation dataset to the height estimation formula created from the training dataset (Fig. 7, S1 Table). Most residuals between estimated values based on lumbar vertebra length and actual height showed a negative correlation with age (R = −0.375~−0.249), though this correlation was weaker compared to residuals based on sternum length (R = −0.474) [32]. Among those where $R^2$ was high in the above analysis, PLV1 exhibited a weak negative correlation with age (R = −0.292, p = 0.009), whereas PLV2, SPLV, and LVTL showed no correlation with age. Therefore, height estimation formulas based on PLV2, SPLV, and LVTL were found to be unaffected by age.

### Discussion

All measured lengths of the lumbar spine examined in this study had a significant positive correlation with height as indicated by regression analysis, suggesting that this parameter is suitable for height estimation. In particular, the LVTL, SPLV, and PLV2 all had $R^2$ exceeding 0.6 and an SEE of about 4.5 cm, indicating the possibility of highly accurate height estimation. The results of regression analysis between residuals and age showed that these three measured lengths are

**Table 5. Difference (residual) between the actual height and the estimated values derived from each estimation formula (Analysis of Assumptions in Linear Regression).**

|  | Shapiro-Wilk test p-value |
|---|---|
| ALV1 | 0.779 |
| CLV1 | 0.348 |
| PLV1 | 0.523 |
| ALV2 | 0.354 |
| CLV2 | 0.559 |
| PLV2 | 0.367 |
| ALV3 | 0.711 |
| CLV3 | 0.537 |
| PLV3 | 0.761 |
| ALV4 | 0.255 |
| CLV4 | 0.174 |
| PLV4 | 0.712 |
| ALV5 | 0.937 |
| CLV5 | 0.255 |
| PLV5 | 0.984 |
| SALV | 0.164 |
| SCLV | 0.362 |
| SPLV | 0.831 |
| LVTL | 0.208 |
| SB | 0.718 |

ALV, anterior margin height of lumber vertebra; CLV, central height of lumber vertebra; PLV, posterior margin height of lumber vertebra; SALV, the sum of ALV; SCLV, the sum of CLV; SPLV, the sum of PLV; LVTL, the lumbar vertebra total length; SB, sternal bone. The specific lumbar vertebrate number is indicated in each dataset measurement.

not affected by aging. Validation tests also proved to be an appropriate estimation formula, as the ME of LVTL, SPLV, and PLV2 were approximately ±1.0 cm and the MAE was < 4.5 cm. The $R^2$ for the sternal bone (SB) was extremely smaller (0.389) and the SEE was larger (6.119 cm) than that of the estimation formula based on the three lumbar spine lengths. Furthermore, the results of regression analysis between residuals and age revealed that the height estimation equation based on sternal length was affected by aging, and was judged to be less suitable for height estimation than the lengths of the lumbar spine measurements. According to previous reports [12,25–27], the $R^2$ of height estimation based on sternal length ranged from 0.153 to 0.517 (median, 0.212) and SEE from 4.76 to 6.73 (median, 5.92) cm, which are similar to our results. However, both are less accurate than our formula for height estimation based on the measured length of the lumbar spine.

Studies on height estimation based on the measured lengths of lumbar vertebrae, as in the present study, have been reported in the past, some based on actual measurements at autopsy and others based on measurements on postmortem images (CT and MRI), as in the present study [28]. In Zhang *et al*. [28] with the highest $R^2$ (0.629) among the previous studies, $R^2$ is considered to be large because the estimation equation was created by dividing samples by age: 25–45 years old and 45 years old or older. In actual forensic practice, the bodies requiring height estimation are unidentified and of unknown age, and, thus, the practical application of the formula of Zhang *et al*. is considered difficult. In Nagesh *et al*. [14] with the next highest $R^2$ (male, 0.434; female, 0.389), the subjects were in their 20s to 50s (mean age for males,

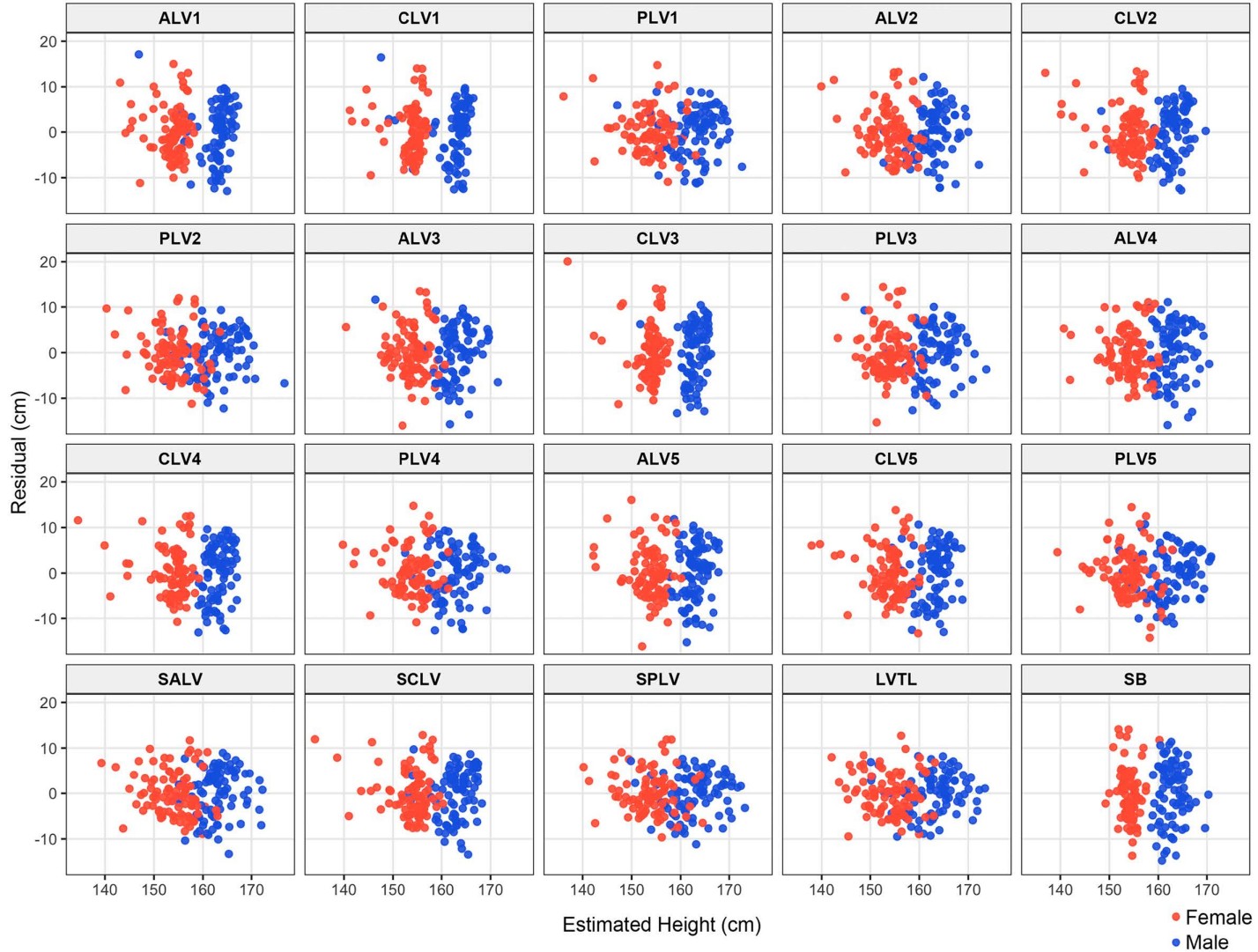

**Fig 5. Scatter plot showing the relationship between residuals and estimated heights.** All residuals are independent and exhibit constant homogeneity of variance. ALV, anterior margin height of lumber vertebra; CLV, central height of lumber vertebra; PLV, posterior margin height of lumber vertebra; SALV, the sum of ALV; SCLV, the sum of CLV; SPLV, the sum of PLV; LVTL, the lumbar vertebra total length; SB, sternal bone. The specific lumbar vertebrate number is indicated in each dataset measurement.

33.44±8.67 years; mean age for females, 27.47±7.07 years)–clearly younger than our study sample (19–95 years old). The $R^2$ is considered to be larger in the younger age group because age-related changes such as compression fractures are not observed in this group. However, it is clear that the estimation equations developed using such samples do not take into account the effects of aging, and, thus, cannot accurately estimate height in the elderly. Jason *et al*. [33] also included subjects in a similar age range (18–86 years) as those in our study, but they excluded cases with compression fractures, so the effect of aging was not considered and $R^2$ was 0.288 (American white women) to 0.471 (American black women), which is smaller than that in our results. Finally, Terazawa *et al*. [34] also included Japanese subjects, but the age range of their sample was relatively young, up to 60 years old, and the number of cases studied was quite small (42 males and 29 females), which reduced the correlation coefficients to 0.283 for males and 0.194 for females. Since none

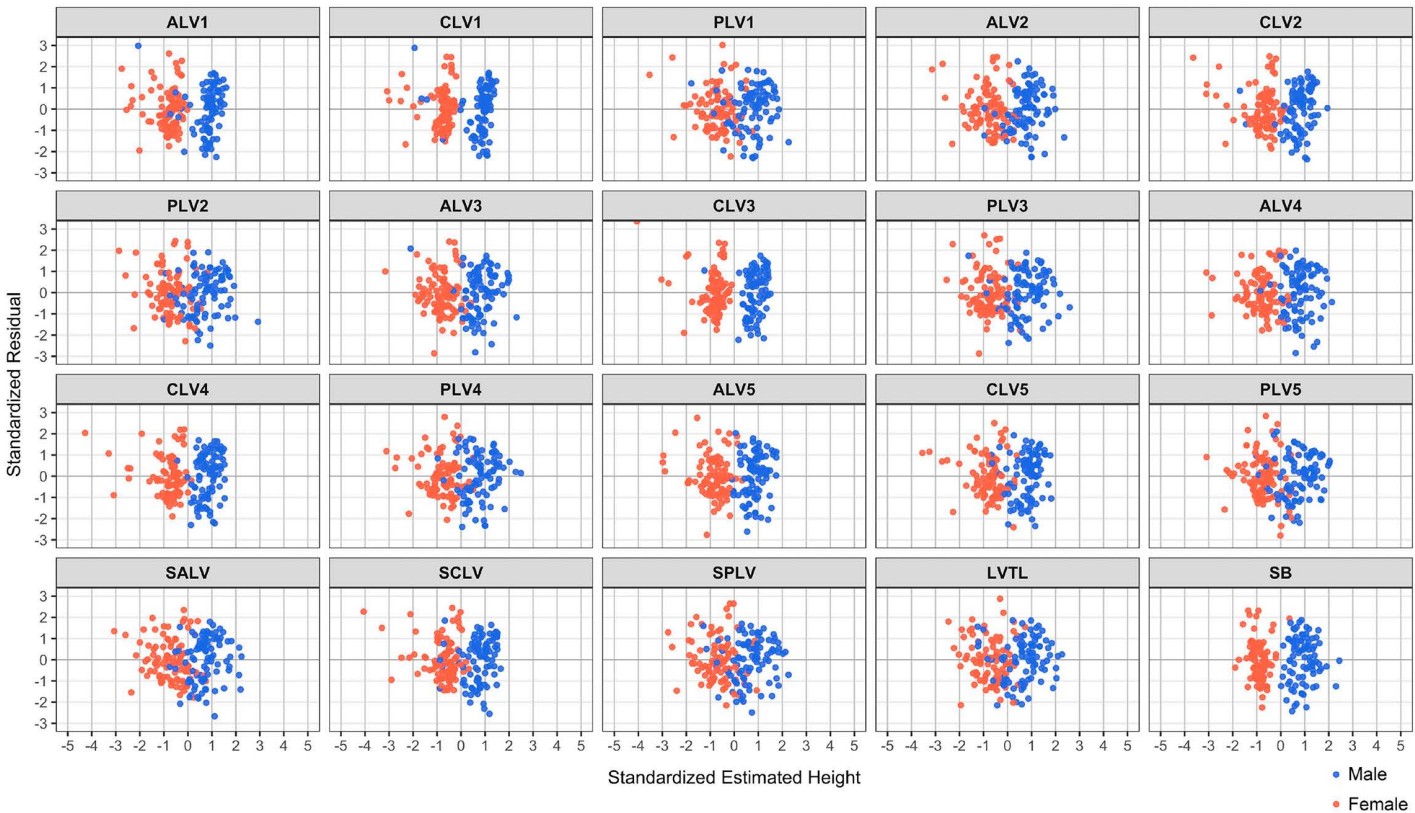

**Fig 6. Scatter plot showing the relationship between standardized residuals and standardized estimated heights.** No specific pattern is observed across all residuals, indicating linearity. ALV, anterior margin height of lumber vertebra; CLV, central height of lumber vertebra; PLV, posterior margin height of lumber vertebra; SALV, the sum of ALV; SCLV, the sum of CLV; SPLV, the sum of PLV; LVTL, the lumbar vertebra total length; SB, sternal bone. The specific lumbar vertebrate number is indicated in each dataset measurement.

of the previous reports exceeded our results in accuracy of height estimation and did not take into account the effects of age-related changes, we believe that our formula is superior to those developed previously. In addition, previous reports excluded cases with compression fractures due to their influence on the formulation of the height estimation formula. In order to include the effect of age-related height shortening in the formulation of the new height estimation equation, we included cases with old compression fractures (35.5% of all training dataset cases) and scoliosis (7.1% of all training dataset cases) in the study. The proportion of cases with pathological conditions such as compression fractures and scoliosis that should be included in order to create a more accurate height estimation equation is a subject for future study.

In comparing the measured lengths of each lumbar vertebra, the correlation was stronger for PLV than for ALV or CLV in all cases. The reason for this may be attributed to the deformation pattern of the lumbar vertebrae due to changes in vertebral body shape with aging and vertebral compression fractures [23]. Vertebral compression fractures are classified into fish, wedge, and flat vertebrae according to the shape of the fracture, all of which have marked shortening of the anterior margin height and central height, and only mild shortening of the posterior margin height [23,35]. Therefore, the shortening of the anterior border height and the median height is an age-related change, but individual differences are large, while the shortening of the posterior border height shows less individual change but definite shortening, and, thus, is considered to appropriately reflect age-related changes. For all lumbar vertebrae except ALV4, a higher correlation with height was observed in males than in females (Table 3). In general, women have a higher frequency of pathological

**Table 6. Mean error (bias) and mean absolute error (inaccuracy) results of validation test.**

|  | ME (cm) | MAE (cm) |
|---|---|---|
| ALV1 | −0.581 | 4.098 |
| CLV1 | −0.561 | 4.020 |
| PLV1 | −0.753 | 3.734 |
| ALV2 | −1.071 | 4.413 |
| CLV2 | −1.253 | 4.030 |
| PLV2 | −1.277 | 4.066 |
| ALV3 | −1.161 | 4.070 |
| CLV3 | −0.918 | 4.096 |
| PLV3 | −0.908 | 3.991 |
| ALV4 | −0.622 | 4.207 |
| CLV4 | −0.485 | 3.846 |
| PLV4 | −0.567 | 3.811 |
| ALV5 | −0.194 | 4.095 |
| CLV5 | −0.703 | 4.065 |
| PLV5 | −0.467 | 4.315 |
| SALV | −0.569 | 3.656 |
| SCLV | −0.712 | 3.653 |
| SPLV | −0.747 | 3.819 |
| LVTL | 0.016 | 3.712 |
| SB | −1.015 | 4.444 |

ME, mean error; MAE, mean absolute error; ALV, anterior margin height of lumber vertebra; CLV, central height of lumber vertebra; PLV, posterior margin height of lumber vertebra; SALV, the sum of ALV; SCLV, the sum of CLV; SPLV, the sum of PLV; LVTL, the lumbar vertebra total length; SB, sternal bone. The specific lumbar vertebrate number is indicated in each dataset measurement.

conditions of the lumbar vertebrae, such as compression fractures due to osteoporosis, and the correlation is considered to be lower because of the tendency for individual differences to occur. Therefore, we believe that the height estimation formula based on the sum of the height of the posterior margin is useful.

Finally, in the present study, a sagittal section image of the lumbar vertebrae was created based on the 3D constructed CT image, and a height estimation equation was created from the values obtained by measuring the length of each lumbar vertebrae. This could also be applied using simple X-ray images (sagittal section) of the lumbar spine. As radiography has been common in the forensic practice for a long time, it is likely that radiography equipment is available. Therefore, this estimation formula can be applied widely in forensic practice.

## Limitations

There are limitations to this study.

1. LVTL cannot be measured in cadavers without remaining intervertebral discs. On the other hand, SPLV and PLV2 can be measured even in cadavers of bones only, but SPLV can only be measured if all of the first to fifth lumbar vertebrae remain. PLV2 can only be measured if the second lumbar vertebrae remain, making it impossible to estimate height.

2. This study is based on a limited sample of CT images of deceased cases in one prefecture where the authors reside. Since it is clear that there are ethnic differences in height [1], future studies with samples from other ethnic groups are needed to generalize the height estimation formula developed in this study.

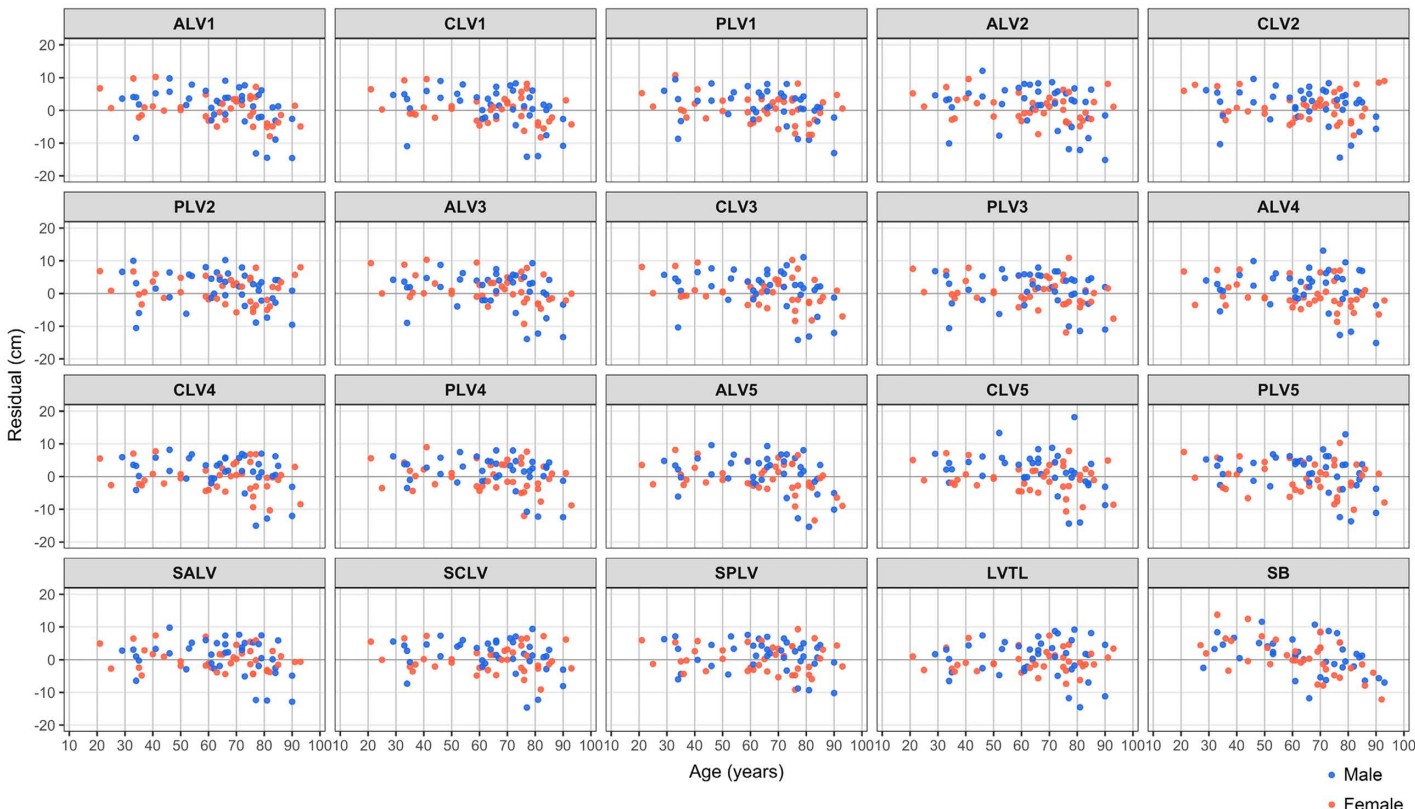

**Fig 7. Scatter plot showing the relationship between each residual and age.** ALV, anterior margin height of lumber vertebra; CLV, central height of lumber vertebra; PLV, posterior margin height of lumber vertebra; SALV, the sum of ALV; SCLV, the sum of CLV; SPLV, the sum of PLV; LVTL, the lumbar vertebra total length; SB, sternal bone. The specific lumbar vertebrate number is indicated in each dataset measurement.

3. This study used postmortem CT images as samples, and the correct height was measured from the top of the head to the bottom of the feet in the supine position on the autopsy table, which was measured at the time of autopsy, and is different from the method used to measure height in living persons in addition to influence of postmortem changes in tissue. Therefore, the height estimation formula developed in this study cannot be directly applied to a living person.

4. Although there were few intra- or inter-measurer errors in this study, the measurement of lumbar vertebrae length on CT images can lead to measurement errors depending on the anatomical variations or pathologies/deformities of the corpse, which may affect the estimation of height.

## Conclusions

In conclusion, we believe that our formula for estimating height based on lumbar vertebrae measurements is suitable regardless of body shape, which has been a problem in the past. In addition, our formula uses bones that are likely to remain as indicators after burnout, damage, or destruction and takes into account the effects of aging, which have not been examined in previous reports. However, the formula should be updated periodically to take into account changes in human body shape over time.

## Supporting information

**S1 File. Anonymized data for lumber spine length measurement.**
(XLSX)

**S2 File. Anonymized data for sternum length measurement.**
(XLSX)

**S1 Table. Results of correlation analysis between residuals derived from each height estimation formula and age.**
(DOCX)

## Acknowledgments

We sincerely thank Prof. Hirofumi Nakaoka (Department of Data Science, Graduate School of Medical and Dental Sciences, Kagoshima University, Kagoshima Japan) for his advice on statistical analysis of the revise submission.

## Author contributions

**Conceptualization:** Takahito Hayashi.

**Data curation:** Takahito Hayashi.

**Formal analysis:** Kotomi Kai, Midori Katsuyama.

**Investigation:** Kotomi Kai, Midori Katsuyama, Takahito Hayashi.

**Methodology:** Kotomi Kai.

**Project administration:** Takahito Hayashi.

**Supervision:** Takahito Hayashi.

**Visualization:** Midori Katsuyama.

**Writing – original draft:** Kotomi Kai.

**Writing – review & editing:** Midori Katsuyama, Takahito Hayashi.

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
