## [Decision Letter · Decision Letter 0]

29 Jan 2025

PONE-D-24-54653Novel height estimation formula that accounts for the effects of aging based on lumbar length measurements in postmortem CT imagesPLOS ONE

Dear Dr. Hayashi,

Thank you for submitting your manuscript to PLOS ONE. After careful consideration, we feel that it has merit but does not fully meet PLOS ONE’s publication criteria as it currently stands. Therefore, we invite you to submit a revised version of the manuscript that addresses the points raised during the review process.

**ACADEMIC EDITOR:** 

Dear Dr. Takahito Hayashi,

We appreciate you submitting your manuscript to PLOS ONE and thank you for giving us the opportunity to consider your work.

I have completed my evaluation of your manuscript, which has been reviewed by two highly qualified reviewers all of whom agree it is worth to be published in PLOS ONE. Nevertheless, they have suggested some minor changes that will help to improve the paper.

Therefore, I invite you to resubmit your manuscript after addressing the reviewers’ comments below. When revising your manuscript, please consider all issues mentioned in the reviewers' comments carefully: please, outline every change made in response to their comments and provide suitable rebuttals for any comments not addressed. Please, note that your revised submission may need to be re-reviewed.

PLOS ONE values your contribution and I look forward to receiving your revised manuscript.

Yours sincerely,

Dr. Olga Spekker

We look forward to receiving your revised manuscript.

Kind regards,

Olga Spekker, Ph.D.

Academic Editor

PLOS ONE

Journal Requirements:

“All authors declare no conflict of interest related to this study.”

Reviewers' comments:

Reviewer's Responses to Questions

**Comments to the Author**

1. Is the manuscript technically sound, and do the data support the conclusions?

Reviewer #1: Partly

Reviewer #2: Yes

2. Has the statistical analysis been performed appropriately and rigorously? 

Reviewer #1: N/A

Reviewer #2: Yes

3. Have the authors made all data underlying the findings in their manuscript fully available?

Reviewer #1: Yes

Reviewer #2: Yes

4. Is the manuscript presented in an intelligible fashion and written in standard English?

Reviewer #1: No

Reviewer #2: Yes

5. Review Comments to the Author

Reviewer #1: The Advantages of Study:

a. Addresses Age-Related Height Loss: The study recognises and attempts to address the problem of overestimation in height prediction for older people, a common problem with traditional method based on limb bone length. This is a significant advantage as it aims to improve accuracy specifically for an age group where existing methods for all short.

b. Uses Lumbar Spine Measurements: The study focus on lumbar spine measurements offers a practical alternative to limb bone measurements. Lumbar vertebrae are more likely to remain intact in situations where limb bones might be missing or damaged (e.g., fire victims, severe trauma). This makes the proposed method potentially applicable in forensic contexts where traditional methods are unusable.

c. Contemporary of Data set: the study uses a contemporary data set (2016-2023) of postmortem CT images. This is crucial for developing accurate estimation formulas that reflect current body proportions and account for secular changes in stature. Older formulas based on historical populations may not be as accurate for modern individuals.

d. Age-related spinal changes: The study explicitly considers age-related changes in the, such as decreased bone density, compression fractures, and osteophyte formation. Incorporating these factors into the height estimation formula is a key strength. Contributing to its potential for improved accuracy in older individuals.

e. Sternal length: the inclusion of sternal length as a control provides a valuable comparison point. This allows for an assessment of the lumbar spine methods performance relative to another skeletal element from the trunk region.

f. Ethics: The study has an ethics statement that ensure the research conducted in accordance with local legislation and institutional requirements, which helps to ensure the ethical treatment of human participants.

Study Limitations:

a. Limited generalization: the study population is drawn from a specific geographic location (the author’s institute). This raises questions about the generalizability of the findings to other populations with different body proportions and age-related changes in stature. Further validation in diverse populations is needed.

b. Sample: the study relies on postmortem CT images, while this provides a large sample size, it is important to consider potential differences between living and deceases individuals. Factors like postmortem changes in tissue and posture could introduce some bias into the measurements.

c. Lack of focus on Lumbar Spine: while the study focus on the lumbar spine is advantageous in some contexts, it also presents a limitation. Accurate measurement of the lumbar vertebrae requires CT imaging, which may not always be readily available. This limits the applicability of the method compared to simpler techniques using easily measurable skeletal elements.

d. Measurement: Measuring vertebral heights and lengths from CT images can be subject to measurement error, particularly if image quality is suboptimal or if there are anatomical variations or pathologies present. The study does not explicitly address to potential impact of measurement error on the accuracy of the height e

Reviewer #2: The manuscript introduces a reliable method for estimating height in deceased individuals, especially the elderly, by using lumbar spine measurements. It accounts for age-related spinal shortening, improving height estimates for older individuals. The study demonstrates robust correlations between lumbar spine indices and height, offering a more accurate alternative to traditional methods like sternal length for older bodies. Additionally, the inclusion of cadavers across a wide age range enhances the relevance of the findings, making the method valuable for forensic investigations, particularly in cases involving elderly individuals.

i have few points for revision of the manuscript

It would be helpful to clarify the time interval between death and CT imaging, as postmortem decomposition can affect spinal measurements and height estimation.

The study should specify whether postmortem CT is routinely used or only in specific cases. This will help assess the method's applicability in standard forensic investigations.

Including more representative CT images would strengthen the methodology section, showcasing how measurements were taken across different age groups and cadavers.

The study focuses only on the lumbar spine, but long bones like the femur and tibia also contribute to height. Including these measurements might improve the accuracy, especially in fragmented bodies.

It’s important to mention whether radiographs were also performed. Since X-rays are more commonly used in forensic practice, it would be helpful to know if the formula can be applied to radiographs.

Incorporating measurements from long bones like the femur and tibia could enhance the reliability of height estimation, especially when the spine is fragmented or poorly preserved.

The formula could be validated with data from living populations or a combination of living and postmortem data to ensure its generalizability across both groups.

The impact of pathological conditions (e.g., fractures, scoliosis, osteoarthritis) on spinal height should be considered. Either excluding individuals with these conditions or accounting for them could make the findings more robust.

Further analysis of gender differences in lumbar spine measurements would help refine the height estimation formula for both males and females.

6. PLOS authors have the option to publish the peer review history of their article (what does this mean? ). If published, this will include your full peer review and any attached files.

**Do you want your identity to be public for this peer review?** For information about this choice, including consent withdrawal, please see our Privacy Policy .

Reviewer #1: No

Reviewer #2: **Yes:** Ishan Kumar

---

## [Author Response · Author response to Decision Letter 1]

13 Mar 2025

Dear Editor and Reviewers,

We thank you for your critical and instructive comments. We have read the comments carefully and revised the manuscript in response. In the revised manuscripts, we have highlighted the modified points.

Comments to Reviewer #1:

Thank you for your comments on our manuscript. We appreciate your understanding of the advantages and importance of our research.

Answers to Reviewer #1:

1. Limited generalization: the study population is drawn from a specific geographic location (the author’s institute). This raises questions about the generalizability of the findings to other populations with different body proportions and age-related changes in stature. Further validation in diverse populations is needed.

As you point out, the study is based on a limited sample of CT images of deceased cases in our prefecture. It is clear that there are ethnic differences in height, and, therefore, we have mentioned this point in the Limitations section that similar studies using samples from other ethnic groups are needed to generalize the present height estimation formula in the future. (please see page 18, lines 336-339 of Limitations and new references [33]).

2. Sample: the study relies on postmortem CT images, while this provides a large sample size, it is important to consider potential differences between living and deceases individuals. Factors like postmortem changes in tissue and posture could introduce some bias into the measurement

This study used postmortem CT images as samples, and the correct height was measured at the time of autopsy from the top of the head to the bottom of the feet in the supine position on the autopsy table, and is different from the method used to measure height in living persons, in addition to the influence of postmortem changes in tissue. Therefore, the height estimation formula developed in this study cannot be applied directly to a living person. Accordingly, we have added a mention about this in the Limitations section. (please see pages 18, lines 340-345 of Limitations).

3. Lack of focus on Lumbar Spine: while the study focus on the lumbar spine is advantageous in some contexts, it also presents a limitation. Accurate measurement of the lumbar vertebrae requires CT imaging, which may not always be readily available. This limits the applicability of the method compared to simpler techniques using easily measurable skeletal elements.

As you pointed out, postmortem CT images of the lumbar vertebrae were used to estimate height based on the equations obtained in this study, and CT images are not always readily available. This limits the practical application of this method compared to other methods using easily measurable bone, such as the humerus, radius, or ulna. Accordingly, we have discussed this in the Limitations section. (please see page 18, lines 346-349 of Limitations).

4. Measurement: Measuring vertebral heights and lengths from CT images can be subject to measurement error, particularly if image quality is suboptimal or if there are anatomical variations or pathologies present. The study does not explicitly address to potential impact of measurement error on the accuracy of the height.

In accordance with your suggestion, the measurement of lumbar vertebrae length on CT images can potentially lead to measurement errors depending on the anatomical variations or pathologies/deformities of the corpse, which may affect the estimation of height. Accordingly, we have mentioned this in the Limitations section. (please see pages 18-19, lines 350-353 of Limitations).

Comments to Reviewer #2:

Thank you for your comments on our manuscript. We appreciate your understanding of the advantages and importance of our research.

Answers to Reviewer #2:

1. It would be helpful to clarify the time interval between death and CT imaging, as postmortem decomposition can affect spinal measurements and height estimation.

As you suggested, we added to the time interval between death and CT imaging in the Materials and Methods section. The average time interval was 4.8 days (range, 12 hours to 60 days; about 90% of cases within 7 days). Cases in which post-mortem decomposition clearly affected the reading were excluded from the sample. (please see pages 5, lines 112-114 of Materials).

2. The study should specify whether postmortem CT is routinely used or only in specific cases. This will help assess the method's applicability in standard forensic investigations.

In response to your suggestion, we have added text describing that, at our department, postmortem CT imaging are routinely performed in all autopsy cases except for completely skeletonized cadavers, and includes 200-250 cases per year. (please see page 5, lines 101-103 of Materials).

3. Including more representative CT images would strengthen the methodology section, showcasing how measurements were taken across different age groups and cadavers.

We agree with your suggestion and have added to Figure 2 a case with osteophyte formation and a case with old lumbar compression fracture in the lumbar vertebral body, which is difficult to measure. (please see new Figure 2 and its caption in page 6, lines 129-138 of Measurement of lumber spine length of Results).

4. The study focuses only on the lumbar spine, but long bones like the femur and tibia also contribute to height. Including these measurements might improve the accuracy, especially in fragmented bodies. Incorporating measurements from long bones like the femur and tibia could enhance the reliability of height estimation, especially when the spine is fragmented or poorly preserved.

Thank you for your comment. Due to the specifications of the CT equipment at our institution, the imaging range was from the head to the pelvic region, and it was not possible to measure lower limb bones such as the femur and tibia. On the other hand, as a preliminary study for this study, we attempted to create a height estimation formula based on the maximum length of the radius and ulna, which have traditionally been measured for height estimation in forensic practice, and we confirmed that in both cases, the estimated height tended to be estimated greater than actual height in elderly people. Accordingly, we have added text in the Introduction section and a scatterplot showing the difference between height estimates based on each bone length and the actual height as Supplemental Figure 1. (please see page 4, lines 74-76 of Introduction, Supplemental Figure 1 and its caption in page 4, lines 89-93).

5. It’s important to mention whether radiographs were also performed. Since X-rays are more commonly used in forensic practice, it would be helpful to know if the formula can be applied to radiographs.

In the present study, a sagittal section image of the lumbar vertebrae was created based on the 3D constructed CT image, and a height estimation equation was created from the values obtained by measuring the height of each lumbar vertebrae. This could also be applied using simple X-ray images (sagittal section) of the lumbar spine. As radiography has been common in the forensic practice for a long time, it is likely that radiography equipment is available even without CT equipment. Therefore, this estimation formula, which is thought to be able to estimate height even with simple X-ray images, can be applied widely in forensic practice. Accordingly, we have added this to the Discussion. (please see page 17, lines 322-328 of Discussion).

6. The formula could be validated with data from living populations or a combination of living and postmortem data to ensure its generalizability across both groups.

This study used postmortem CT images as samples, and the correct height was measured at the time of autopsy from the top of the head to the bottom of the feet in the supine position on the autopsy table. This method is different from that used to measure height in living persons, and accounts for the influence of postmortem changes in tissue. Therefore, the height estimation formula developed in this study cannot be applied directly to a living person. Accordingly, we have mentioned this in the Limitations section. (please see pages 18, lines 340-345 of Limitations).

7. The impact of pathological conditions (e.g., fractures, scoliosis, osteoarthritis) on spinal height should be considered. Either excluding individuals with these conditions or accounting for them could make the findings more robust.

Thank you for pointing this out. We focused on age-related changes in lumbar spine length, such as old lumbar vertebral compression fractures and lumbar scoliosis, and conducted this study in the belief that measuring lumbar spine length may provide an accurate height estimate that takes age-related changes into account. The original submission did not include the frequencies of old compression fractures and scoliosis in the sample, and, therefore, we have added these variables in the revised version (please see page 5, lines 108-112 of Materials). Moreover, we have added a new note in the Discussion stating that adding those cases to the sample has allowed us to develop an accurate height estimation equation that takes into account age-related height shortening. (please see page 16, lines 300-304 of Discussion).

8. Further analysis of gender differences in lumbar spine measurements would help refine the height estimation formula for both males and females.

Thank you for your comment. We initially created the height estimation equation separately for males and females. As a result, we found that for most of the measured lengths in this study, the correlation with height is greater when male and female data are combined than when these data are separated. However, we also found that for AVL1, AVL2, AVL3, and SAVL, the correlation with height was greater when male and female data were separated than when they were combined. We have added text to the Discussion to address this point. (please see page 17, lines 314-319 of Discussion).

Other modifications:

We extended the period during which we had access to the CT images due to the need to review the post-mortem CT images again in order to respond to the reviewers' comments (please see page 5, lines 101-102). Accordingly, we also applied to our institutional ethics committee to extend the duration of the study, and, as the extension was approved, we have attached the ethics review approval forms (original and translated versions) for the study extension. (please see page 7, line 158 and attached files named by “original ethics approval letter 2” and “approval document 2 (Translation)”).

We thank the reviewers for their comments and feel that the revised manuscript is much improved. We hope that you find the manuscript suitable for publication in PLos One. We thank you in advance for your kindness and look forward to your favorable reply.

With my best regards,

---

## [Decision Letter · Decision Letter 1]

5 May 2025

PONE-D-24-54653R1Novel height estimation formula that accounts for the effects of aging based on lumbar length measurements in postmortem CT imagesPLOS ONE

Dear Dr. Hayashi,

Thank you for submitting your manuscript to PLOS ONE. After careful consideration, we feel that it has merit but does not fully meet PLOS ONE’s publication criteria as it currently stands. Therefore, we invite you to submit a revised version of the manuscript that addresses the points raised during the review process.

We look forward to receiving your revised manuscript.

Kind regards,

Olga Spekker, Ph.D.

Academic Editor

PLOS ONE

**Additional Editor Comments:**

Dear Dr. Takahito Hayashi,

We appreciate you submitting your manuscript to PLOS ONE and thank you for giving us the opportunity to consider your work.

I have completed my evaluation of your revised manuscript, which has been reviewed by two highly qualified reviewers. Unfortunately, Reviewer 4 suggested rejection of your manuscript. Reviewer 3 also highlighted substantial issues but would like to give a chance for rebuttal. Therefore, I recommend major revision of your manuscript.

Both reviewers mentioned issues with data availability; therefore, it is advised to upload your data to a trusted data repository as the reviewers suggested. Furthermore, as Reviewer 3 highlighted, the lumbar measurements in the supplementary data file may be wrong (maybe they got mixed up before submission) as while they tried to reproduce the analysis with the data you provided, they were able to match the summary statistics exactly (Table 1), but “none of the correlation coefficients or linear models of the lumbar measurements matched the presented results (Tables 2, 3, and 4). And it was quite a large difference, to the point where it would completely change the interpretation.”. Based on the results of Reviewer 3, the stature estimation method you propose will not work as there is very little correlation between any of your variables and the known height. Unfortunately, if the provided lumbar measurement data have not been mixed up before submission and not your but Reviewer 3’s findings are correct, I will have to reject your manuscript. Therefore, I kindly ask you to provide an explanation for what could have caused the difference between your and Reviewer 3’s results.

Based on the above, I invite you to resubmit your manuscript after addressing the reviewers’ comments below. When revising your manuscript, please consider all issues mentioned in the reviewers' comments carefully: please, outline every change made in response to their comments and provide suitable rebuttals for any comments not addressed. Please, note that your revised submission may need to be re-reviewed.

PLOS ONE values your contribution and I look forward to receiving your revised manuscript.

Yours sincerely,

Dr. Olga Spekker

Reviewers' comments:

Reviewer's Responses to Questions

**Comments to the Author**

1. If the authors have adequately addressed your comments raised in a previous round of review and you feel that this manuscript is now acceptable for publication, you may indicate that here to bypass the “Comments to the Author” section, enter your conflict of interest statement in the “Confidential to Editor” section, and submit your "Accept" recommendation.

Reviewer #3: (No Response)

Reviewer #4: All comments have been addressed

2. Is the manuscript technically sound, and do the data support the conclusions?

Reviewer #3: Partly

Reviewer #4: Yes

3. Has the statistical analysis been performed appropriately and rigorously? 

Reviewer #3: No

Reviewer #4: Yes

4. Have the authors made all data underlying the findings in their manuscript fully available?

Reviewer #3: Yes

Reviewer #4: No

5. Is the manuscript presented in an intelligible fashion and written in standard English?

Reviewer #3: Yes

Reviewer #4: Yes

6. Review Comments to the Author

Reviewer #3: # Notes about the manuscript

l. 74-76 & Supplemental Figure 1 - this seems a little awkward and should probably be moved to the results or completely removed from the manuscript, since it's not really the focus of the study. I see that it was added as a response to a reviewer, but I really don't think it's necessary to include more variables. If you do decide to keep this, make sure you also add the appropriate information to the Materials and Methods section (i.e., the information about how the measurements were taken, etc), and the results should be added to the tables (but I really think it's best just to remove it).

Table 1 - it's not clear why Age is presented twice, and why the bottom three rows only have 244 individuals. I'm guessing that this is because the last three rows represent the sample associated with the sternal measurements, which is different from the sample used for lumbar measurements. If so, this should be more clear in the table/paper.

l.235-236 & 237-238 - the p-value is not a direct probability of significance, but rather the probability of observing the data given the null hypothesis is true.

Table 4 - I'm more interested in the actual estimates (coefficients) of the regression model than the p-values since these are more indicative of the effect of age, so these should be presented as well. The table should also present the multiple R-squared value, not the correlation coefficient (R).

Model evaluation: Given that this study is assessing the prediction ability of the model, it would be better to highlight the standard error, which is more indicative of prediction accuracy than the r-squared. Even better model evaluation would be to separate the data into training and prediction sets, and/or to use some form of cross-validation. Without this kind of evaluation it's not really possible to talk about prediction accuracy. It's also important to address the assumptions of linear regression. Are the residuals roughly normally distributed, is there independence of observations, do the residuals have constant variance (homoscedasticity), and is there a linear relationship between the predictor and response variables?

Data availability - There is a discrepancy between the data availability statement in the manuscript and the statement in the metadata. The former states that data are available upon request (not sufficient according to PLOS Data Policy), while the latter states the data are fully available without restriction, and are available in the manuscript and supporting information (which seems to be the case). It would in any case be better to upload the data to a trusted data repository (like Zenodo or an institutional repository) to adhere to the FAIR principles. There also needs to be more documentation associated with the data file, like a README file or a data dictionary. The data in the file should also be more rectangular to facilitate re-use. There are currently many levels of headers, such as 'Lumbar length' > 'AVL1' > 'Measured value'. Such information about the data should be presented in a separate file (like the aforementioned README). Also, instead of 'Measured value', consider using the name of the actual measurement, e.g., ALV1.

I tried to reproduce the analysis using the data provided in the supplementary materials (*minimal data set.xlsx*) and the methods described in the Methods section. I was able to match the summary statistics in Table 1, as well as the correlation coefficients and linear models for the sternal measurements (SB). However, none of the correlation coefficients or linear models of the lumbar measurements matched the presented results (Tables 2, 3, and 4). And it was quite a large difference, to the point where it would completely change the interpretation. Based on the results I got, the stature estimation method you propose will not work as there is little-to-no correlation between any of the variables and the height. Is it possible that the lumbar measurements in the data file are wrong (maybe they got mixed up before they were submitted)?

I have attached a zip file (reprex.zip) with the R script I used to reproduce some of the results.

I would like to give a chance for rebuttal, so I am recommending Major Revision of the manuscript; this is a valuable study and I would like to see it published given that the methodology of data collection seems sound, and it addresses a gap in the available data for the region. To be published, the statistical analysis and presentation will need a significant overhaul, and the associated data will need to be corrected.

Reviewer #4: (No Response)

7. PLOS authors have the option to publish the peer review history of their article (what does this mean? ). If published, this will include your full peer review and any attached files.

**Do you want your identity to be public for this peer review?** For information about this choice, including consent withdrawal, please see our Privacy Policy .

Reviewer #3: **Yes:** Bjørn Peare Bartholdy

Reviewer #4: No

---

## [Author Response · Author response to Decision Letter 2]

31 Oct 2025

Dear Editor and Reviewers,

We thank you for your critical and instructive comments. We have read the comments carefully, revised the manuscript in response, and highlighted the modified points.

Answers to Reviewer #1:

1. 74-76 & Supplemental Figure 1 - this seems a little awkward and should probably be moved to the results or completely removed from the manuscript, since it's not really the focus of the study. I see that it was added as a response to a reviewer, but I really don't think it's necessary to include more variables. If you do decide to keep this, make sure you also add the appropriate information to the Materials and Methods section (i.e., the information about how the measurements were taken, etc), and the results should be added to the tables (but I really think it's best just to remove it).

As you suggested, the relevant text and Supplemental Figure 1 have been removed from the revised manuscript (see second paragraph of Introduction).

2. Table 1 - it's not clear why Age is presented twice, and why the bottom three rows only have 244 individuals. I'm guessing that this is because the last three rows represent the sample associated with the sternal measurements, which is different from the sample used for lumbar measurements. If so, this should be clearer in the table/paper.

As you pointed out, the sample sizes used for lumbar spine length measurement and sternum length measurement differ. Therefore, to clarify the text for the readers, the revised version separates the list of samples used for lumbar spine length measurement into Table 1 and the list of samples used for sternum length measurement into Table 2. Additionally, the reason for the differing sample sizes has been added to the text (see Tables 1 and 2, pages 5-6, lines 118-120 of Materials).

3. l.235-236 & 237-238 - the p-value is not a direct probability of significance, but rather the probability of observing the data given the null hypothesis is true.

We have deleted the relevant passage and revised it with a different expression (see pages 19-20, lines 338-346 of Results).

4. Table 4 - I'm more interested in the actual estimates (coefficients) of the regression model than the p-values since these are more indicative of the effect of age, so these should be presented as well. The table should also present the multiple R-squared value, not the correlation coefficient (R).

As you suggested, we included the R-squared value instead of the R value in Table 4 (see Table 4).

5. Model evaluation: Given that this study is assessing the prediction ability of the model, it would be better to highlight the standard error, which is more indicative of prediction accuracy than the r-squared. Even better model evaluation would be to separate the data into training and prediction sets, and/or to use some form of cross-validation. Without this kind of evaluation it's not really possible to talk about prediction accuracy. It's also important to address the assumptions of linear regression. Are the residuals roughly normally distributed, is there independence of observations, do the residuals have constant variance (homoscedasticity), and is there a linear relationship between the predictor and response variables?

As suggested, we emphasized not only a high R² value but also a low SEE to evaluate the predictive capability of the model we created (see page 8, lines 179-181 of Statistical Analysis, pages 13-14, lines 250-258 of Height estimation formula, page 22, lines 369-375 of Discussion). Additionally, to ensure proper model evaluation, we split the dataset into training and validation sets to assess predictive accuracy (see pages 5-6, lines 102-127 of Materials, page 8, lines 183-185 of Statistical analysis, Figure 1, Tables 1 and 2, page 18, lines 307-312 of Validation test, page 22, lines 372-375 of Discussion). Further, to address the assumptions of linear regression (see page 8, lines 174-179 of Statistical analysis, page 14, lines 258-261 of Height estimation formula), we analyzed whether the residuals (1) are normally distributed (see Table 5), (2) are independent (see Table 5), (3) have constant variance (homoscedasticity) (see Figure 5), and (4) exhibit linearity between standardized residuals and standardized predicted values. (see Figure 6). To perform statistical analyses more efficiently, the statistical software was changed from Excel Statistics to R (see pages 7, lines 163-164 of Statistical analysis).

6. Data availability - There is a discrepancy between the data availability statement in the manuscript and the statement in the metadata. The former states that data are available upon request (not sufficient according to PLOS Data Policy), while the latter states the data are fully available without restriction, and are available in the manuscript and supporting information (which seems to be the case). It would in any case be better to upload the data to a trusted data repository (like Zenodo or an institutional repository) to adhere to the FAIR principles. There also needs to be more documentation associated with the data file, like a README file or a data dictionary. The data in the file should also be more rectangular to facilitate re-use. There are currently many levels of headers, such as 'Lumbar length' > 'AVL1' > 'Measured value'. Such information about the data should be presented in a separate file (like the aforementioned README). Also, instead of 'Measured value', consider using the name of the actual measurement, e.g., ALV1.

As suggested, we have uploaded the sample data used in this study to a trusted data repository (https://zenodo.org/records/17430849). We organized the data as suggested and also uploaded a README file (see page 27, lines 481-482 of Data availability).

7. I tried to reproduce the analysis using the data provided in the supplementary materials (*minimal data set.xlsx*) and the methods described in the Methods section. I was able to match the summary statistics in Table 1, as well as the correlation coefficients and linear models for the sternal measurements (SB). However, none of the correlation coefficients or linear models of the lumbar measurements matched the presented results (Tables 2, 3, and 4). And it was quite a large difference, to the point where it would completely change the interpretation. Based on the results I got, the stature estimation method you propose will not work as there is little-to-no correlation between any of the variables and the height. Is it possible that the lumbar measurements in the data file are wrong (maybe they got mixed up before they were submitted)?

We sincerely apologize. The data uploaded previously contain errors. We have confirmed that all analyses are reproducible before revising the submission and have uploaded the correct data.

Other modifications:

As you can see in the newly added Figure 3 in this revised version, the height distributions for men and women are clearly different. Therefore, whereas previous versions created separate height estimation equations for men and women from the outset, this revised version incorporates gender as an explanatory variable to create a single height estimation equation using combined data for both sexes (see pages 7-8, lines 166-172 and Figure 3). Additionally, following the switch to R for statistical analysis software, Figures 1, 3–7 were recreated using R instead of Excel software (see page 7, line 163-164 and Figures 1, 3-7).

We thank the reviewers for their comments and feel that the revised manuscript is much improved. We hope that you find the manuscript suitable for publication in PLoS One. We thank you in advance for your kindness and look forward to your favorable reply.

With my best regards,

---

## [Decision Letter · Decision Letter 2]

18 Nov 2025

PONE-D-24-54653R2Novel height estimation formula that accounts for the effects of aging based on lumbar length measurements in postmortem CT imagesPLOS ONE

Dear Dr. Hayashi,

Thank you for submitting your manuscript to PLOS ONE. After careful consideration, we feel that it has merit but does not fully meet PLOS ONE’s publication criteria as it currently stands. Therefore, we invite you to submit a revised version of the manuscript that addresses the points raised during the review process.

**ACADEMIC EDITOR:**

Dear Dr. Takahito Hayashi,

We appreciate you submitting your manuscript to PLOS ONE and thank you for giving us the opportunity to consider your work.

I have completed my evaluation of your manuscript, which has been reviewed by the same reviewer who evaluated it in the second round. They agree it is worth to be published in PLOS ONE. Nevertheless, they have suggested some minor changes that will help to improve the paper.

Therefore, I invite you to resubmit your manuscript after addressing the reviewer’s comments below. When revising your manuscript, please consider all issues mentioned in the reviewer's comments carefully: please, outline every change made in response to their comments and provide suitable rebuttals for any comments not addressed. Please, note that your revised submission may need to be re-reviewed.

PLOS ONE values your contribution and I look forward to receiving your revised manuscript.

Yours sincerely,

Dr. Olga Spekker

We look forward to receiving your revised manuscript.

Kind regards,

Olga Spekker, Ph.D.

Academic Editor

PLOS ONE

Journal Requirements:

Reviewers' comments:

Reviewer's Responses to Questions

**Comments to the Author**

1. If the authors have adequately addressed your comments raised in a previous round of review and you feel that this manuscript is now acceptable for publication, you may indicate that here to bypass the “Comments to the Author” section, enter your conflict of interest statement in the “Confidential to Editor” section, and submit your "Accept" recommendation.

Reviewer #3: All comments have been addressed

2. Is the manuscript technically sound, and do the data support the conclusions?

Reviewer #3: Yes

3. Has the statistical analysis been performed appropriately and rigorously? 

Reviewer #3: Yes

4. Have the authors made all data underlying the findings in their manuscript fully available?

Reviewer #3: Yes

5. Is the manuscript presented in an intelligible fashion and written in standard English?

Reviewer #3: Yes

6. Review Comments to the Author

Reviewer #3: Data - great that they are now available on Zenodo, and in a form that is much easier to work with. I would recommend having all the raw data together in a single sheet in addition to the sheets containing the separated training and validation subsets.

Analysis - since the revised analysis was conducted in R, it would be great if the R script could also be shared on the Zenodo repository. Note: if you create a new version of the Zenodo repository, remember to update the link (DOI) in the manuscript.

Results - Still some discrepancies in the correlation coefficients, but within a 5% margin which is reasonable.

l. 341 & l. 345 - It seems a little arbitrary to state that -0.292 is a very weak negative correlation while -0.474 is a significant negative correlation. Unless you have another rule-of-thumb you can refer to, the former can probably be described as a small correlation and the latter as a medium correlation (according to Jacob Cohen's recommendation, 1992).

Table 7 - This table is a bit confusing. The caption says regression analysis but the table presents a correlation coefficient. I'm guessing the caption should state 'Results of correlation analysis...' While there are no strong correlations between the measurements and Age, it may be worth noting that they are all negatively correlated (as would be expected from degenerative changes related to age), so the age estimation equation is consistently overestimating height as age increases, but to a lesser extent than sternal length. This table can probably also be moved to the supplementary materials.

R should be added to the references: R Core Team (2025). R: A Language and Environment for Statistical Computing. R Foundation for Statistical Computing, Vienna, Austria. https://www.R-project.org/.

Cohen, J. (1992). A power primer. Psychological Bulletin, 112(1), 155–159. https://doi.org/10.1037/0033-2909.112.1.155

7. PLOS authors have the option to publish the peer review history of their article (what does this mean? ). If published, this will include your full peer review and any attached files.

**Do you want your identity to be public for this peer review?** For information about this choice, including consent withdrawal, please see our Privacy Policy .

Reviewer #3: **Yes:** Bjørn Peare Bartholdy

---

## [Author Response · Author response to Decision Letter 3]

25 Nov 2025

Dear Reviewers,

We thank you for your critical and instructive comments. We have read the comments carefully, revised the manuscript in response, and highlighted the modified points.

Answers to Reviewer #3:

1. Data - great that they are now available on Zenodo, and in a form that is much easier to work with. I would recommend having all the raw data together in a single sheet in addition to the sheets containing the separated training and validation subsets.

We are glad that you were able to successfully use the data we uploaded to Zenodo repository. As you pointed out, we uploaded the training subset and validation subset as separate files, each on a single sheet. Additionally, within the Folder containing the uploaded the R script, we have also included Text files created from the same Excel files (see Zenodo repository: https://doi.org/10.5281/zenodo.17677851).

2. Analysis - since the revised analysis was conducted in R, it would be great if the R script could also be shared on the Zenodo repository. Note: if you create a new version of the Zenodo repository, remember to update the link (DOI) in the manuscript.

We have uploaded all R scripts used for statistical analysis to the above Zenodo repository (a new version of the Zenodo repository). The files containing the R scripts were compressed into Folder (ZIP format) for upload. To reproduce the analyses, please download the entire Folder (named CT_project) and extract its contents. When using R, please select this Folder as your Working Directory in R Studio. Please use the “main_script” located in the folder named “scripts” within the same Folder (see Zenodo repository: https://doi.org/10.5281/zenodo.17677851).

3. Results - Still some discrepancies in the correlation coefficients, but within a 5% margin which is reasonable.

As you pointed out, after redoing all statistical analyses, we found minor errors in some Tables and have corrected them (see Tables 3, 6 and Supplemental Table 1).

4. l. 341 & l. 345 - It seems a little arbitrary to state that -0.292 is a very weak negative correlation while -0.474 is a significant negative correlation. Unless you have another rule-of-thumb you can refer to, the former can probably be described as a small correlation and the latter as a medium correlation (according to Jacob Cohen's recommendation, 1992).

As you pointed out, our interpretation of the strength of the correlation was somewhat arbitrary. Therefore, following the advice in the reference you suggested, we have revised the description (see page 20, lines 339-343 of Results). We have also added that reference to Reference lists as [32] (see new Reference [32]).

5. Table 7 - This table is a bit confusing. The caption says regression analysis but the table presents a correlation coefficient. I'm guessing the caption should state 'Results of correlation analysis...' While there are no strong correlations between the measurements and Age, it may be worth noting that they are all negatively correlated (as would be expected from degenerative changes related to age), so the age estimation equation is consistently overestimating height as age increases, but to a lesser extent than sternal length. This table can probably also be moved to the supplementary materials.

As you pointed out, the caption for Table 7 was incorrect since it represents a correlation analysis rather than a regression analysis. I have corrected it accordingly (see page 21, the caption of Supplemental Table 1). Furthermore, the explanation of the results in Table 7 was confusing, so we have revised them to appropriate descriptions (see pages 19-20, lines 336-343 of Results). Additionally, following your advice, we have moved Table 7 to Supplemental Table 1. (see page 21, Supplemental Table 1).

5. R should be added to the references: R Core Team (2025). R: A Language and Environment for Statistical Computing. R Foundation for Statistical Computing, Vienna, Austria. https://www.R-project.org/.

As you advised, we have added references on analysis using R (see page 7, lines 164 of Statistical analysis and new Reference [29]).

Other modifications:

In the residual analysis of regression analysis, we performed the Durbin-Watson test for testing independence in the manuscript we submitted last time. However, we found that this test method was incorrect because our research sample is not time-series data. We assessed independence (presence or absence of autocorrelation) by examining the scatterplot of residuals versus estimated height (Figure 5) and confirming the absence of obvious patterns. Consequently, we revised part of the methods (see page 8, lines 175-177 of Statistical Analysis), part of the results (see page 14, line 258 of Results), and Table 5 (see page 16).

We thank the reviewers for their comments and feel that the revised manuscript is much improved. We hope that you find the manuscript suitable for publication in PLoS One. We thank you in advance for your kindness and look forward to your favorable reply.

With my best regards,

---

## [Decision Letter · Decision Letter 3]

2 Dec 2025

Novel height estimation formula that accounts for the effects of aging based on lumbar length measurements in postmortem CT images

PONE-D-24-54653R3

Dear Dr. Hayashi,

We’re pleased to inform you that your manuscript has been judged scientifically suitable for publication and will be formally accepted for publication once it meets all outstanding technical requirements.

Kind regards,

Olga Spekker, Ph.D.

Academic Editor

PLOS ONE

Additional Editor Comments (optional):

Reviewers' comments:

Reviewer's Responses to Questions

**Comments to the Author**

1. If the authors have adequately addressed your comments raised in a previous round of review and you feel that this manuscript is now acceptable for publication, you may indicate that here to bypass the “Comments to the Author” section, enter your conflict of interest statement in the “Confidential to Editor” section, and submit your "Accept" recommendation.

Reviewer #3: All comments have been addressed

2. Is the manuscript technically sound, and do the data support the conclusions?

Reviewer #3: Yes

3. Has the statistical analysis been performed appropriately and rigorously? 

Reviewer #3: Yes

4. Have the authors made all data underlying the findings in their manuscript fully available?

Reviewer #3: Yes

5. Is the manuscript presented in an intelligible fashion and written in standard English?

Reviewer #3: Yes

6. Review Comments to the Author

Reviewer #3: (No Response)

7. PLOS authors have the option to publish the peer review history of their article (what does this mean? ). If published, this will include your full peer review and any attached files.

**Do you want your identity to be public for this peer review?** For information about this choice, including consent withdrawal, please see our Privacy Policy .

Reviewer #3: **Yes:** Bjørn Peare Bartholdy

---

## [Editor Report · Acceptance letter]

PONE-D-24-54653R3

PLOS One

Dear Dr. Hayashi,

I'm pleased to inform you that your manuscript has been deemed suitable for publication in PLOS One. Congratulations! Your manuscript is now being handed over to our production team.

Kind regards,

on behalf of

Dr. Olga Spekker

Academic Editor

PLOS One